# Aerobic exercise interventions to address impaired quality of life in patients with pituitary tumors

Christopher S. Hong[1,2]*, Timothy R. Smith[1,2]

1 Department of Neurosurgery, Brigham and Women's Hospital, Harvard Medical School, Boston, Massachusetts, United States of America, 2 Computational Neuroscience Outcomes Center (CNOC), Boston, Masachusettts, United States of America

* chong6@mgb.org

**Data Availability Statement:** The article does not report data and the data availability policy is not applicable.

## Abstract

Patients with pituitary tumors may experience persistent fatigue and reduced physical activity, based on subjective measures after treatment. These symptoms may persist despite gross total resection of their tumors and biochemical normalization of pituitary function. While reduced quality of life has been commonly acknowledged in pituitary tumor patients, there is a lack of studies on what interventions may be best implemented to ameliorate these issues, particularly when hormonal levels have otherwise normalized. Aerobic exercise programs have been previously described to ameliorate symptoms of chronic fatigue and reduced physical capacity across a variety of pathologies in the literature. As such, a prescribed aerobic exercise program may be an underrecognized but potentially impactful intervention to address quality of life in pituitary tumor patients. This review seeks to summarize the existing literature on aerobic exercise interventions in patients with pituitary tumors. In addition, future areas of study are discussed, including tailoring exercise programs to the hormonal status of the patient and incorporating more objective measures in monitoring response to interventions.

## Introduction

Patients with pituitary tumors can suffer from a variety of symptoms due to the many complex functions of the pituitary gland. While there are clear cut indications for surgical and/or medical intervention in biochemically proven cases of hormonal excess or deficiency, as well as visual compromise, patients often report more subjective complaints related to an overall diminished quality of life, citing factors ranging from undue fatigue, increased anxiety, depression, poor sleep, and worsened cognition. In many cases, despite normalization of the hormonal axis through surgery and/or a medical prescription, patients continue to experience decreased quality of life parameters, compared to healthy individuals. For example, patients with Cushing's disease may continue to experience excessive fatigue and diminished cognition, despite normalization of cortisol levels [1]. Patients with acromegaly may still suffer from increased anxiety and personal image perception even after treatment [2]. Even in patients

**Funding:** The author(s) received no specific funding for this work.

**Competing interests:** The authors have declared that no competing interests exist.

with pituitary adenomas for whom physical functioning is less affected compared to those with acromegaly and Cushing's disease, feelings of undue fatigue as well as mental disorders and poor sleep have been reported [3–5].

Aerobic exercise is well-accepted in the literature to reduce the burden of many health problems. There are innumerable studies with well-constructed aerobic exercise interventions resulting in improved physical performance, cognitive well-being, and overall quality of life in various disease states, ranging from dementia, cancer, and arthritis among others [6–9]. However, despite the clear vulnerability that a diagnosis of a pituitary adenoma has on the hormonal function of a patient, there are relatively few studies analyzing the impact of aerobic exercise interventions in improving overall quality of life in patients with pituitary adenomas. In particular, such intervention may have the potential to address more subjective measures of quality of life in these patients who continue to suffer despite adequate biochemical normalization of their hormones with surgical and/or medical treatment.

This review seeks to overview the literature on aerobic exercise interventions in patients with pituitary adenomas. In addition, we discuss future directions in this arena, including salient points that relate to personalizing aerobic exercise interventions for patients with specific pituitary-related disorders, as well as combining objective measures along with existing subjective measures to monitor response to treatment.

## Methods

A literature search was performed utilizing the PubMed index and was queried with the following search terms: "aerobic exercise", "physical exercise", "pituitary", "sellar", "suprasellar". Exclusion criteria included studies of patients with absent pituitary or sellar pathologies and a lack of an implemented aerobic exercise intervention. After an initial search result of 1963 studies, 225 were further reviewed, based on screening the titles. Subsequently, abstract review of these studies resulted in 4 studies that were included in this review (Fig 1). Specifically, these articles were included, based on the fact they pertained to patients with a diagnosis of pituitary

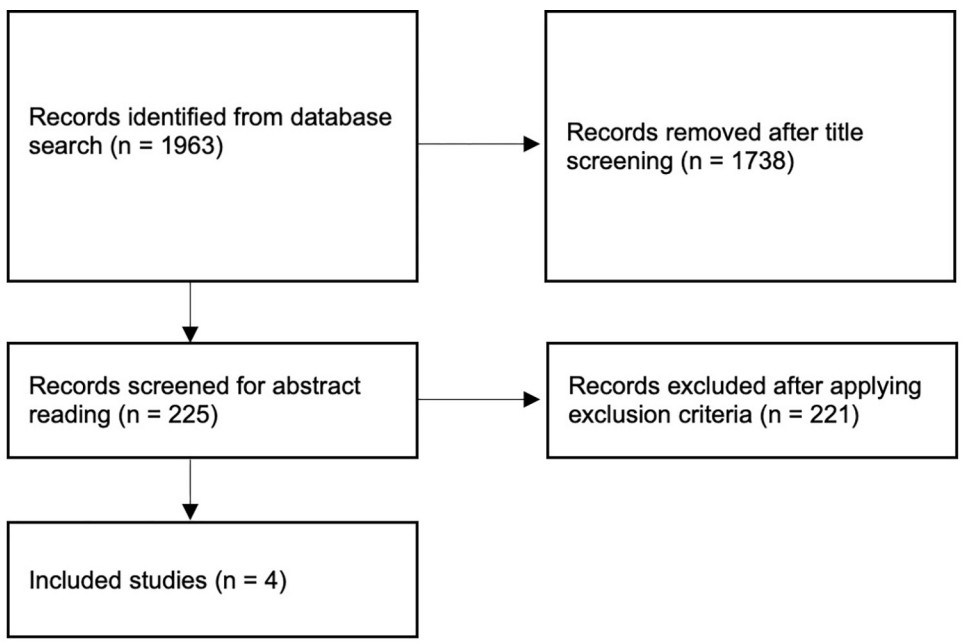

**Fig 1. Flow diagram depicting search strategy for literature review.**

adenoma or a sellar/suprasellar tumor, a specific prescribed aerobic exercise intervention, and reported outcomes after said intervention. Articles that were included were written in English and published as recently as August 2023.

## Results

### Impaired quality of life in pituitary tumor patients

Patients with pituitary tumors frequently experience difficulties with quality of life related to physical and mental fatigue, despite normal hormonal levels. Two prospective studies of patients treated for pituitary tumors and biochemical normalization of hormonal levels still demonstrated reduced quality of life metrics and undue fatigue, based on validated health-related questionnaires (Sickness Impact Profile [10], Hospital Anxiety and Depression Scale [11], Multidimensional Fatigue Index [12], Nottingham Health Profile [13], Shorter Form-36 [14]) [15, 16]. Notably, there are data to suggest that increased physical activity after pituitary tumor surgery may improve fatigue level scores in these patients. Zhao *et al.* performed a prospective, observational study of 184 patients undergoing pituitary adenoma resection and found that patients who scored higher on self-driven increased physical activity via the International Physical Activity questionnaire [17] after surgery correlated with reduced levels of reported fatigue [18].

In addition, several studies have reported lasting sleep disturbances despite complete pituitary adenoma resection and normalization of pituitary function. Two groups reported persistent sleep impairment as measured with the Epworth Sleepiness Scale [19], Berlin Questionnaire [20], and Pittsburgh Sleep Quality Index [21], despite improvement in other quality of life metrics as measured with the Shorter Form-26, Hospital Anxiety and Depression Scale, Multidimensional Fatigue Index, M.D. Anderson Symptom Inventory Brain Tumor Module [22], and European Organization for Research and Treatment of Cancer QLQ-C30 [23]) in patients after pituitary surgery, including objective measures of sleep quality with polysomnography and actigraphy [24, 25]. Furthermore, patients treated for pituitary macroadenomas or craniopharygniomas showed increased daytime somnolence despite normal sleeping patterns [26, 27]. These patients may also benefit from aerobic exercise interventions, which have been shown in meta-analyses to positively impact disrupted sleep patterns in patients with chronic insomnia, as well as the elderly population as a whole [28, 29]. No studies have been performed to date to analyze the impact of aerobic exercise on improvement of sleep quality after pituitary tumor surgery.

There is also evidence for disease-specific quality of life impairments in patients with different pituitary tumor subtypes. For example, patients with acromegaly and Cushing's disease have reported persistent impairment of physical functioning and bodily pain, despite demonstrating biochemical normalization, as measured via the Hospital Anxiety and Depression Scale, Multidimensional Fatigue Index, Nottingham Health Profile, and Shorter Form-36 questionnaires [30]. In addition, patients with Cushing's disease may suffer from long-term cognitive impairments, as measured via the Fatigue impact scale [31], and reduced quality of life, evaluated through the Shorter Form-36 questionnaire, even though they are in remission [1]. Another study reported that female patients treated for microprolactinomas still experienced increased fatigue, decreased physical activity, and high levels of anxiety and depression, as measured with the Shorter Form-36, Nottingham Health Profile, Multidimensional Fatigue Index, and Hospital Anxiety and Depression Scale questionnaires [32].

Although craniopharyngiomas are distinct from pituitary tumors, their sellar/suprasellar location can lead to pituitary dysfunction. In addition, hypothalamic injury can occur after surgical treatment leading to devastating metabolic consequences, including severe obesity,

sleep disturbance, thermal dysregulation, among others [33]. However, even in cases of maximally safe surgical resection and biochemical normalization of pituitary function, several studies have demonstrated decreased activity levels and impaired aerobic adaptation to physical exercise [34–36]. While there have been many attempts to study various pharmacologic interventions to address the metabolic disturbances seen in craniopharyngioma patients with hypothalamic injury, there are a lack of data to address impaired quality of life in patients treated for craniopharyngioma with preserved hypothalamic function [37, 38]. Given the evidence for decreased physical functioning, a tailored aerobic exercise intervention could be effective in optimizing activity levels and overall quality of life for these patients.

Taken together, there is evidence of a persistent quality of life reduction in patients with pituitary tumors, despite treatment and biochemical normalization of hormonal abnormalities. While aerobic exercise interventions have been widely studied across a variety of diseases to address quality of life metrics, there are relatively few data in the pituitary tumor population.

**Studies on aerobic exercise interventions in pituitary tumor patients.** Dulger *et al*. performed a prospective interventional study on aerobic exercise and yoga intervention in patients with pituitary tumors [39]. Specifically, participants underwent aerobic exercise training on a treadmill for 30 minutes, three days a week for six weeks. Each session was followed by strength training, comprised of calisthenics and weight training. A yoga training program was also implemented, performed three days a week for 60 minutes a day. The program was specifically designed to avoid movements or poses that might be higher risk for patients with pituitary adenomas and their associated comorbidities of muscle atrophy and bone density. The study enrolled 10 patients, all of whom had undergone surgery for pituitary macroadenomas a mean 6.2 years prior and at the time of the study had no clinical signs of hypopituitarism. Patients were randomized into either the combined aerobic and strength program or the yoga program for six weeks, followed by a washout period of two weeks, and then undergoing the other intervention for six weeks. Outcomes measured included various quality of life metrics evaluating sleep, fatigue, anxiety, depression, sexual function, and cognitive function. The yoga program resulted in significant increases in FACT-Br score (an overall quality of life metric in brain cancer patients), the aerobic and strength training program resulted in significantly lower anxiety scores, and both interventions led to significantly increased cognition scores. In addition, the authors found non-significant trends of decreased fatigue scores after the aerobic and strength program and decreased fatigue, anxiety, and depression scores and increased sexual function after the yoga program. Taken together, this study demonstrated that a combination of yoga and a combined aerobic and strength training program led to improved overall quality of life in patients with pituitary adenomas.

Lima *et al*. conducted a prospective interventional study on 17 patients with acromegaly, the majority of whom had controlled disease through a combination of surgical and/or medical therapies [40]. Patients underwent an exercise program guided by a physiotherapist of strength, aerobic, and flexibility exercises, lasting 60 minutes and conducted three times a week for two months. Outcomes measured included the Acromegaly Quality of Life (AcroQoL) questionnaire, a general fatigue questionnaire used to assess patients with chronic illness, and multiple physical fitness assessments including handgrip strength, lower limb muscle strength, static body balance, and walk test. These evaluations were performed after completion of the exercises at two months, and again one month later to serve as a washout period. After completion of the exercise intervention, there were significant improvements observed in measures of general fatigue, lower limb muscle strength, the walk test, balance control, and overall quality of life. However, after the one-month washout period, only the balance control and lower extremity strength improvements remained statistically significant. These findings

demonstrated the potential of aerobic exercise as a meaningful short-term intervention for patients with acromegaly while also highlighting the need for longer-term implementation to contribute towards improving overall quality of life as an adjunctive treatment for patients with biochemically controlled acromegaly.

Hatipoglu *et al.* conducted a prospective case-control study of aerobic exercise intervention in 11 patients with acromegaly, compared to nine age and gender matched acromegalic patients in a control group [41]. All patients had achieved biochemical normalization of their pituitary hormonal axis through a combination of medical and/or surgical treatment. The aerobic exercise intervention consisted of a supervised 75 minute session comprised of cardiac, strength, and balance and stretching exercises, performed three days a week for three consecutive months. Outcomes measured included an acromegaly specific quality of life questionnaire, a depression score, and a body image questionnaire. At the completion of the study, there were no significant differences in body mass index (BMI), growth hormone (GH), and insulin-like growth factor 1 (IGF-1) levels although BMI and IGF-1 levels tended to decline in the interventional cohort. Likewise, there were no changes in the depression and quality of life scores. However, there was a significant improvement in the body image questionnaire in patients undergoing the aerobic exercise intervention, compared to controls, despite no changes in BMI or biochemical markers of GH secretion. In a separate study, the authors also reported more objective measures of response to the exercise intervention, including significant improvements in exercise tolerance, as measured by maximal oxygen consumption and a graded exercise test, as well as decreases in body fat [42].

As such, these studies demonstrated that exercise intervention may improve overall body image and perception of patients with acromegaly, independent from the effects on body composition and disease activity. In addition, there may be objective improvement of physical functioning, which may beneficially address the detrimental effects on functional performance from the cardiovascular changes seen in acromegaly [43, 44].

The aforementioned studies are summarized in Table 1.

## Discussion

Although our review of the literature found very few studies reporting aerobic exercise interventions in patients with pituitary tumors, these studies demonstrated such an intervention may lead to improvements in a patient population that is already predisposed to impaired quality of life. That said, it remains unclear why patients treated for pituitary tumors and with normalization of their hormones still exhibit persistent physical fatigue [25]. Some authors have hypothesized that this may be due to sleep disturbances, while others have suggested a self-perpetuating cycle of low physical activity, related to a modern sedentary lifestyle, compounded by prescribed reductions in physical exertion after surgery, that may exacerbate subjective feelings of fatigue [25, 45, 46].

Indeed, a recent study by Zhao *et al.* in 184 patients undergoing pituitary adenoma resection found that fatigue and physical activity levels were significantly affected immediately after surgery but improved over the course of several months after surgery [18]. Patients undergoing transsphenoidal resection of their tumors may be predisposed to higher levels of fatigue after surgery, as they are typically instructed to avoid strenuous physical activity to maintain the integrity of skull base repair [47]. However, graded exercise interventions in patients after major surgery or experiencing chronic fatigue may lead to improvements in fatigue levels [48, 49]. A specific and tailored aerobic exercise program may be a viable interventional strategy in pituitary tumor patients after surgery to avoid falling into long-term ailment from a myriad of physical and psychosocial symptoms.

**Table 1. Summary of literature review of studies on aerobic exercise interventions in pituitary tumor patients.**

| Authors (reference) | Study population | Study design | Intervention | Outcome measures | Results |
|---|---|---|---|---|---|
| Dulger et al, 2022 [39] | 10 patients with history of surgically resected pituitary macroadenomas, no clinical signs of hypopituitarism | Randomized crossover | Treadmill exercise (30 minutes) folowed by strength training for 3 days/week x 6 weeks Yoga program (60 minutes) for 3 days/week x 6 weeks | • FACT-Br (quality of life)<br>• Pittsburgh Sleep Quality Index<br>• Fatigue Severity Scale (FSS)<br>• Female Sexual Function Index (FSFI)<br>• Hospital Anxiety and Depression Scale (HADS)<br>• Montreal Cognitive Assessment Scale (MOCA) | • Significantly higher FACT-Br scores and nonsignificant decreases in HADS anxiety scores and increased FSFI scores after yoga program<br>• Significantly lower HADS anxiety scores after treadmill/strength program<br>• Significantly higher MOCA and nonsignificant decrease in FSS scores after both yoga and treadmill/strength programs |
| Lima et al, 2019 [40] | 17 patients with medically/surgically controlled acromegaly | Prospective interventional | Physiotherapist guided strength, aerobic, and flexibility exercises (60 minutes) for 3 days/week x 8 weeks | • Body composition through bioimpedance Acromegaly Quality of Life (AcroQoL)<br>• Functional Assessment of Chronic Illness Therapy<br>• Fatigue scale Handgrip strength<br>• Lower extremity functionality using isometric dynamometer and Lower Extremity Functional Scale (LEFS)<br>• Body balance through stabilometry<br>• Functional capacity through 6-minute walking distance (6MWD) | • Significant improvements in general fatigue, quadriceps muscle strength, LEFS, 6MWD, balance control, and AcroQoL<br>• After 1 month washout period, only significant improvements in LEFS and balance control persisted |
| Hatipoglu et al 2014, [41] | 11 patients in case group with medically/surgically controlled acromegaly 9 patients in control group with medically/surgically controlled acromegaly | Prospective case-control | Cardio, strength, balance and strength session (75 minutes) for 3 days/week x 12 weeks | • Body Mass Index (BMI)<br>• IGF-1 levels<br>• AcroQoL<br>• Beck Depression Inventory (BDI)<br>• Multidimensional Body-Self Relations Questionnaire (MBSRQ) | • Nonsignificant decreases in BMI and IGF-1 levels<br>• Significantly improved MBSRQ (body image) scores |
| Hatipoglu et al 2015, [42] | 11 patients in case group with medically/surgically controlled acromegaly 9 patients in control group with medically/surgically controlled acromegaly | Prospective case-control | Cardio, strength, balance and strength session (75 minutes) for 3 days/week x 12 weeks | • Maximal oxygen consumption (VO2max)<br>• Time to complete Bruce protocol (graded treadmill test)<br>• Muscle flexibility by sit and reach test (SRT)<br>• Muscle strength by hand grip strength test (HGST)<br>• BMI<br>• Waist-to-hip ratio (WHR)<br>• Skinfold measurements from 8 points<br>• Percentage body fat (PBG)<br>• Fat mass (FM)<br>• Lean body mass (LBM) | • Significantly increased VO2max, time to complete Bruce protocol, SRT<br>• Significantly decreased PBF and FM<br>• Stable sum of skinfolds, BMI, WHR, and LBM |

Ultimately, an aerobic exercise intervention needs to be tailored to the specific individual. First of all, exercise modality can influence hormonal response even in healthy people, therefore exercise type, frequency and intensity should be thoughtfully considered in patients with pituitary disease. Endurance exercise, high-intensity interval training, and resistance training

all quite significantly and variably impact the response of growth hormone, prolactin, and cortisol [50]. For patients who have undergone relatively recent pituitary tumor surgery, a lower-intensity regimen of light aerobic exercise in the form of low-impact cardiovascular training, light resistance exercises, and yoga may be more appropriate. On the other hand, for patients with persistent fatigue long after their surgery, higher intensity exercise interventions may be better suited. This post-operative temporal distinction has yet to be explored and reported.

While various questionnaires and scales can be useful to measure quality of life in patients, they are still fraught with inconsistent application and risk of bias. A recent review by van Trigt *et al.* analyzed 20 studies reporting patient-reported outcome measures in patients with refractory hormone-producing adenomas, and found significant variance in conclusions across the studies, compared to patients in biochemical remission [51]. Digital phenotyping has become increasingly utilized, in which data from personal digital devices, in particular smartphones, are gathered to measure behavioral patterns pertaining to sleep, physical mobility, social interactions, and cognitive functioning, among others [52]. This methodology brings a level of objectivity into areas of study that have been reliant on subjective measures, particularly in the field of mental health [53, 54]. Digital phenotyping has been expanded to monitor clinical status in patients suffering from a variety of disease such as long coronavirus disease 2019 (COVID-19), Parkinson's disease, diabetes, asthma, and cardiovascular disease [55–59]. We are currently working to apply this technology to monitor activity levels of patients after pituitary tumor surgery. In addition, it is likely that digital phenotyping may be an additional strategy to measure response to aerobic exercise interventions in pituitary tumor patients, long after their treatment and biochemical normalization of their hormones.

## Conclusions

Taken together, persistent fatigue, reduced activity levels, and overall impaired quality of life remain problematic for patients with pituitary tumors despite adequate treatment. Aerobic exercise interventions have been widely studied across many disease states to alleviate these complaints but remain lacking in the pituitary tumor patient population. The handful of existing studies, as summarized in this review, suggest there is great potential for such interventions to positively impact these patients. However, in patients with pituitary disorders, the aerobic exercise regimen must be tailored to specific individual, taking into consideration the hormonal profile of the pituitary adenoma and the degree to which biochemical control has been achieved. This is particularly relevant for patients with hormone-secreting pituitary tumors, such as Cushing's disease and acromegaly, which can lead to systemic, somatic sequelae that can significantly impair physical function and quality of life. Of particular interest are patients in this category who despite biochemical control of their disease, continue to endorse subjective feelings of undue fatigue and impairment in activites of daily living. Even patients with non-functioning pituitary adenomas and other sellar pathologies such as craniopharyngiomas may suffer similar consequences, perhaps reflective of subclinical detrimental effects on pituitary function. While randomized controlled trials remain the gold standard, further prospective interventional studies are needed to demonstrate the efficacy of aerobic exercise interventions in these patient populations. In addition, future studies should aim to combine disease-specific quality of life measures with objective measures, such as hormone levels and potentially digital phenotyping, in assessing the efficacy of aerobic exercise interventions.

## Author Contributions

**Conceptualization:** Christopher S. Hong, Timothy R. Smith.

**Data curation:** Christopher S. Hong.

**Formal analysis:** Christopher S. Hong, Timothy R. Smith.

**Investigation:** Christopher S. Hong.

**Methodology:** Christopher S. Hong.

**Supervision:** Christopher S. Hong, Timothy R. Smith.

**Validation:** Christopher S. Hong, Timothy R. Smith.

**Visualization:** Christopher S. Hong, Timothy R. Smith.

**Writing – original draft:** Christopher S. Hong.

**Writing – review & editing:** Christopher S. Hong, Timothy R. Smith.

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
