## [Decision Letter · Decision Letter 0]

26 Oct 2023

PONE-D-23-27034Aerobic exercise interventions to address impaired quality of life in patients with pituitary tumorsPLOS ONE

Dear Dr. Hong,

Thank you for submitting your manuscript to PLOS ONE. After careful consideration, we feel that it has merit but does not fully meet PLOS ONE’s publication criteria as it currently stands. Therefore, we invite you to submit a revised version of the manuscript that addresses the points raised during the review process.

We look forward to receiving your revised manuscript.

Kind regards,

Tanja Grubić Kezele, Ph.D., M.D.

Academic Editor

PLOS ONE

Journal Requirements:

**Additional Editor Comments:**

Based on the reviewers' suggestions, the paper needs major revision.  The reviewers' comments can be found below.

Reviewers' comments:

Reviewer's Responses to Questions

**Comments to the Author**

1. Is the manuscript technically sound, and do the data support the conclusions?

Reviewer #1: Yes

Reviewer #2: Yes

2. Has the statistical analysis been performed appropriately and rigorously? 

Reviewer #1: N/A

Reviewer #2: N/A

3. Have the authors made all data underlying the findings in their manuscript fully available?

Reviewer #1: No

Reviewer #2: Yes

4. Is the manuscript presented in an intelligible fashion and written in standard English?

Reviewer #1: Yes

Reviewer #2: Yes

5. Review Comments to the Author

Reviewer #1: Reviewer’s comments

Manuscript number: PONE-D-23-27034

Title: Aerobic exercise interventions to address impaired quality of life in patients with pituitary tumors

Summary

In this review article, the authors summarize previously reported literatures on aerobic exercise interventions in patients with pituitary tumors. This review article looks at what we already know about using exercise to help pituitary tumor patients. It also talks about things we should study in the future, like designing exercise programs based on the patient's hormones and using better ways to see if the exercise is working.

Overall, it's a well-written paper. However, there are some areas that need improvement, such as:

1. More detailed explanations are needed in the 'Method' section, particularly regarding the inclusion and exclusion criteria. Including figures or charts that show the total number of papers searched and the number excluded would be helpful.

2. Summarizing relevant papers related to the topic of this review article in a table would be very useful.

3. Providing more detailed descriptions of areas requiring future research in the conclusion section would be beneficial."

Overall impression

I think that this manuscript contains valuable information worthy of publication in PLOS ONE, after major revision has been made. I would recommend resubmission of the manuscript including considerable revisions.

Thank you very much.

Reviewer #2: The authors have undertaken a literature review on the really interesting topic of fatigue and exercise interventions after pituitary surgery. This is an underrepresented topic in the literature, but a very important quality of life issue. The authors should be commended on undertaking this review and highlighting the problem.

The manuscript is on the whole very well written and a very useful addition to current literature.

Some areas could be improved:

1. The methodology is sparse, only one database was searched. Inclusion and exclusion criteria could be better defined. I accept that the authors were not intending to undertake a systematic review, but the methodology could be much more robust. A table of the papers included would be useful.

2. line 77, validated health related questionnaires - state which

line 79 typo scores fatigue

3. Results section impaired quality of life - this section includes exercise outcomes which should be in the subsequent section. It also lacks detail and is difficult to follow. There should be a better description of the results of the Qol outcome measures in the studies

4. Line 225 - this section of the discussion should be sub headed conclusion

6. PLOS authors have the option to publish the peer review history of their article (what does this mean?). If published, this will include your full peer review and any attached files.

Reviewer #1: No

Reviewer #2: **Yes: **Caroline Hayhurst

---

## [Author Response · Author response to Decision Letter 0]

22 Nov 2023

Additional Editor Comments:

Based on the reviewers' suggestions, the paper needs major revision. The reviewers' comments can be found below.

Response: Thank you for the opportunity to revise our manuscript. Please find below our responses to each reviewers’ suggestions.

Reviewer #1: Reviewer’s comments

Manuscript number: PONE-D-23-27034

Title: Aerobic exercise interventions to address impaired quality of life in patients with pituitary tumors

Summary

In this review article, the authors summarize previously reported literatures on aerobic exercise interventions in patients with pituitary tumors. This review article looks at what we already know about using exercise to help pituitary tumor patients. It also talks about things we should study in the future, like designing exercise programs based on the patient's hormones and using better ways to see if the exercise is working.

Overall, it's a well-written paper. However, there are some areas that need improvement, such as:

1. More detailed explanations are needed in the 'Method' section, particularly regarding the inclusion and exclusion criteria. Including figures or charts that show the total number of papers searched and the number excluded would be helpful.

Response: We have added commentary into the methods on our inclusion and exclusion criteria and also created a flow diagram (figure 1) demonstrating our search strategy for the literature review.

2. Summarizing relevant papers related to the topic of this review article in a table would be very useful. 

Response: We have summarized the relevant papers in newly created Table 1.

3. Providing more detailed descriptions of areas requiring future research in the conclusion section would be beneficial."

Response: We have added additional commentary on specific areas where future research is needed in the conclusion section.

Overall impression

I think that this manuscript contains valuable information worthy of publication in PLOS ONE, after major revision has been made. I would recommend resubmission of the manuscript including considerable revisions.

Thank you very much.

Response: We appreciate the reviewer’s effort and time to evaluate our manuscript. We hope our edits suffice.

Reviewer #2: The authors have undertaken a literature review on the really interesting topic of fatigue and exercise interventions after pituitary surgery. This is an underrepresented topic in the literature, but a very important quality of life issue. The authors should be commended on undertaking this review and highlighting the problem.

The manuscript is on the whole very well written and a very useful addition to current literature.

Some areas could be improved:

1. The methodology is sparse, only one database was searched. Inclusion and exclusion criteria could be better defined. I accept that the authors were not intending to undertake a systematic review, but the methodology could be much more robust. A table of the papers included would be useful.

Response: We have provided a more detailed explanation of our inclusion/exclusion criteria in the Methods section and created a flow diagram, outlining our search strategy (figure 1). Additionally, we have provided a review of the papers in Table 1.

2. line 77, validated health related questionnaires - state which

line 79 typo scores fatigue

Response: We have clarified the health related questionaries in line 77 and provided relevant citations. We have fixed this typo in line 79.

3. Results section impaired quality of life - this section includes exercise outcomes which should be in the subsequent section. It also lacks detail and is difficult to follow. There should be a better description of the results of the Qol outcome measures in the studies

Response: We have edited this section to clarify that the exercises outcomes related to the study by Zhao et al, was an observational study rather than a deliberate interventional study, showing that patients who self reported increased exercise activity after surgery also had lower levels of fatigue. This study differs from those we present in the subsequent section which are specifically describing prospective interventions via exercise regimens and their outcomes. Additionally, we have provided additional details on the methodologies and results on the QoL outcome measures with relevant citations for all of the studies.

4. Line 225 - this section of the discussion should be sub headed conclusion

Response: We have edited this paragraph to be the conclusion section.

---

## [Decision Letter · Decision Letter 1]

4 Dec 2023

Aerobic exercise interventions to address impaired quality of life in patients with pituitary tumors

PONE-D-23-27034R1

Dear Dr. Hong,

We’re pleased to inform you that your manuscript has been judged scientifically suitable for publication and will be formally accepted for publication once it meets all outstanding technical requirements.

Kind regards,

Tanja Grubić Kezele, Ph.D., M.D.

Academic Editor

PLOS ONE

Additional Editor Comments (optional):

Reviewers' comments:

Reviewer's Responses to Questions

**Comments to the Author**

1. If the authors have adequately addressed your comments raised in a previous round of review and you feel that this manuscript is now acceptable for publication, you may indicate that here to bypass the “Comments to the Author” section, enter your conflict of interest statement in the “Confidential to Editor” section, and submit your "Accept" recommendation.

Reviewer #1: All comments have been addressed

Reviewer #2: All comments have been addressed

2. Is the manuscript technically sound, and do the data support the conclusions?

Reviewer #1: Yes

Reviewer #2: (No Response)

3. Has the statistical analysis been performed appropriately and rigorously? 

Reviewer #1: N/A

Reviewer #2: (No Response)

4. Have the authors made all data underlying the findings in their manuscript fully available?

Reviewer #1: Yes

Reviewer #2: (No Response)

5. Is the manuscript presented in an intelligible fashion and written in standard English?

Reviewer #1: Yes

Reviewer #2: (No Response)

6. Review Comments to the Author

Reviewer #1: Dear authors

Thank you very much for your responses.

I’ve read the responses for reviewer’s comments.

Revisions are generally satisfactory.

I feel that this manuscript contains valuable information worthy of publication in Plos One.

Thank you.

Reviewer #2: (No Response)

7. PLOS authors have the option to publish the peer review history of their article (what does this mean?). If published, this will include your full peer review and any attached files.

Reviewer #1: No

Reviewer #2: **Yes: **Caroline Hayhurst

---

## [Editor Report · Acceptance letter]

8 Dec 2023

PONE-D-23-27034R1 

Aerobic exercise interventions to address impaired quality of life in patients with pituitary tumors 

Dear Dr. Hong:

I'm pleased to inform you that your manuscript has been deemed suitable for publication in PLOS ONE. Congratulations! Your manuscript is now with our production department. 

Kind regards, 

on behalf of

Prof. dr. Tanja Grubić Kezele 

Academic Editor

PLOS ONE